# Equivalence of Models in Loop Quantum Cosmology and Group Field Theory

**Bekir Baytaş, Martin Bojowald * and Sean Crowe**

Institute for Gravitation and the Cosmos, The Pennsylvania State University, 104 Davey Lab, University Park, PA 16802, USA; bub188@psu.edu (B.B.); stc151@psu.edu (S.C.)

**\*** Correspondence: bojowald@gravity.psu.edu

**Abstract:** The paradigmatic models often used to highlight cosmological features of loop quantum gravity and group field theory are shown to be equivalent, in the sense that they are different realizations of the same model given by harmonic cosmology. The loop version of harmonic cosmology is a canonical realization, while the group-field version is a bosonic realization. The existence of a large number of bosonic realizations suggests generalizations of models in group field cosmology.

**Keywords:** loop quantum cosmology; group field theory; bosonic realizations

## 1. Introduction

Consider a dynamical system given by a real variable, $V$, and a complex variable, $J$, with Poisson brackets:

$$\{V, J\} = i\delta J \quad , \quad \{V, \bar{J}\} = -i\delta\bar{J} \quad , \quad \{J, \bar{J}\} = 2i\delta V \tag{1}$$

for a fixed real $\delta$. We identify $H_\varphi^\delta = \delta^{-1}\text{Im}J = -i(2\delta)^{-1}(J - \bar{J})$ as the Hamiltonian of the system and interpret $V$ as the volume of a cosmological model. The third (real) variable, $\text{Re}J$, is not independent provided we fix the value of the Casimir $R = V^2 - |J|^2$ of the Lie algebra $su(1,1)$ given by the brackets (1). To be specific, we will choose $R = 0$.

Writing evolution with respect to some parameter $\varphi$, the equations of motion are solved by:

$$V(\varphi) = A\cosh(\delta\varphi) - B\sinh(\delta\varphi) \tag{2}$$

$$\text{Re}J(\varphi) = A\sinh(\delta\varphi) - B\cosh(\delta\varphi). \tag{3}$$

Since $R$ is required to be zero, we obtain $A^2 - B^2 - (\delta H_\varphi^\delta)^2 = 0$, and therefore, there is some $\varphi_0$ such that $A/(\delta H_\varphi^\delta) = \cosh(\delta\varphi_0)$ and $B/(\delta H_\varphi^\delta) = -\sinh(\delta\varphi_0)$. The solution (2) then reads:

$$V(\varphi) = \delta H_\varphi^\delta \cosh(\delta(\varphi - \varphi_0)) \tag{4}$$

and displays the paradigmatic behavior of the volume of a bouncing universe model. This construction defines harmonic cosmology [1,2]. See also [3] for further properties related to $su(1,1)$, in particular group coherent states, and [4] for an application to coarse graining.

The bouncing behavior can also be inferred from an effective Friedmann equation that describes modified evolution of the scale factor giving rise to the volume $V$. To do so, we should provide a physical interpretation to the time parameter $\varphi$ used so far. A temporal description, shared by some models of loop quantum cosmology [5,6] and group field cosmology [7–11], is a so-called internal time [12]: The parameter $\varphi$ is proportional to the value of a scalar field $\phi$ as a specific matter contribution devised such that $\phi$ is in one-to-one correspondence with some time coordinate such

as proper time $\tau$. The scalar $\phi$ itself can then be used as a global time. Its dynamics must be such that its momentum $p_\phi$ never becomes zero; "time" $\phi$ then never stops. With a standard isotropic scalar Hamiltonian:

$$h_\phi = \frac{1}{2}\frac{p_\phi^2}{V} + VW(\phi),$$
(5)

this condition is fulfilled only for vanishing potential $W(\phi)$, such that $p_\phi$ is conserved. The scalar should therefore be massless and without self-interactions. With these conditions, the conserved momentum $p_\phi$ generates "time" translations in $\phi$ and can therefore be identified with the evolution generator $H_\varphi^\delta$ introduced above. In order to match with coefficients in the Friedmann equation derived below, we set:

$$p_\phi = \sqrt{12\pi G} H_\varphi^\delta.$$
(6)

The Hamiltonian (5) also allows us to derive a relationship between $\phi$ and proper time $\tau$, measured by co-moving observers in an isotropic cosmological model. Proper-time equations of motion are determined by Poisson brackets with the Hamiltonian constraint, to which (5) provides the matter contribution. Therefore,

$$\frac{\mathrm{d}\phi}{\mathrm{d}\tau} = \{\phi, h_\phi\} = \frac{p_\phi}{V}.$$
(7)

Writing proper-time derivatives with a dot and using $V = a^3$ to introduce the scale factor $a$, the chain rule then implies:

$$\left(\frac{\dot{a}}{a}\right)^2 = \left(\frac{\dot{\phi}}{3V}\frac{\mathrm{d}V}{\mathrm{d}\phi}\right)^2 = \frac{p_\phi^2}{9V^4}\left(\frac{\mathrm{d}V}{\mathrm{d}\phi}\right)^2$$
(8)

in which:

$$\frac{1}{V^2}\left(\frac{\mathrm{d}V}{\mathrm{d}\phi}\right)^2 = \frac{1}{V^2}\{V, p_\phi\}^2 = 12\pi G\frac{(\mathrm{Re}J)^2}{V^2} = 12\pi G\left(1 - \frac{\delta^2 p_\phi^2}{12\pi G V^2}\right)$$
(9)

follows from the $\phi$-equations of motion, the zero Casimir $R = 0$, and the identification (6) with $H_\varphi^\delta = \delta^{-1}\mathrm{Im}J$. Putting everything together,

$$\left(\frac{\dot{a}}{a}\right)^2 = \frac{4\pi G}{3}\frac{p_\phi^2}{V^2}\left(1 - \frac{\delta^2 p_\phi^2}{12\pi G V^2}\right) = \frac{8\pi G}{3}\rho_\phi\left(1 - \frac{\delta^2 \rho_\phi}{6\pi G}\right)$$
(10)

with the energy density $\rho_\phi = \frac{1}{2}p_\phi^2/a^6$ of the free, massless scalar. Upon rescaling $\delta = 4\pi G\tilde{\delta}$, this effective Friedmann equation agrees with what has been derived in loop quantum cosmology, following [13].

Harmonic cosmology can be obtained as a deformation of a certain model of classical cosmology. In the limit of vanishing $\delta$, $H_\varphi^0 = \lim_{\delta\to 0} H_\varphi^\delta$ has Poisson bracket:

$$\{V, H_\varphi^0\} = \lim_{\delta\to 0}\mathrm{Re}J.$$
(11)

For finite $H_\varphi^0$, we must have $\lim_{\delta\to 0}\mathrm{Im}J = 0$, such that the vanishing Casimir implies $\lim_{\delta\to 0}\mathrm{Re}J = V$. Therefore,

$$\{V, H_\varphi^0\} = V$$
(12)

with an exponential solution $V(\phi) = \exp(\sqrt{12\pi G}\phi)$ that no longer exhibits a bounce. Moreover, noticing that:

$$\{V, V^{-1}H_\varphi^0\} = 1,$$
(13)

we can identify $H_\varphi^0/V = P$ with the momentum canonically conjugate to $V$ in the limit of $\delta \to 0$. Therefore,

$$H_\varphi^0 = VP$$
(14)

is quadratic. Squaring this equation, we find:

$$P^2 = \frac{(H_\varphi^0)^2}{V^2} = \frac{p_\phi^2}{12\pi G V^2} \tag{15}$$

which, upon relating $P = \dot{a}/(4\pi G a)$ to the Hubble parameter and $V$ to the scale factor cubed, is equivalent to the Friedmann equation of an isotropic, spatially-flat model sourced by a free, massless scalar field with momentum $p_\phi$:

$$\left(\frac{\dot{a}}{a}\right)^2 = \frac{8\pi G}{3} \rho_\phi. \tag{16}$$

## 2. Loop Quantum Cosmology as a Canonical Realization of Harmonic Cosmology

It is of interest to construct a canonical momentum $P$ of $V$ also in the case of non-zero $\delta$. The pair $(V, P)$ will then be Darboux coordinates on symplectic leaves of the Poisson manifold defined by (1), and the full (real) three-dimensional manifold will have Casimir–Darboux coordinates $(V, P, R)$. Following the methods of [14], we can construct such a momentum directly from the brackets (1).

Suppose we already know the momentum $P$. The Poisson bracket of any function on our manifold with $V$ then equals the negative derivative by $P$. In particular,

$$\frac{\partial \text{Im} J}{\partial P} = -\{\text{Im} J, V\} = \delta \text{Re} J \tag{17}$$

$$\frac{\partial \text{Re} J}{\partial P} = -\{\text{Re} J, V\} = -\delta \text{Im} J \tag{18}$$

while $\partial V / \partial P = 0$. Up to a crucial sign, these equations are very similar to our equations of motion in the preceding section, and the same is true for their solutions:

$$\text{Im} J(V, P) = A(V) \cos(\delta P) - B(V) \sin(\delta P) \tag{19}$$

$$\text{Re} J(V, P) = -A(V) \sin(\delta P) - B(V) \cos(\delta P). \tag{20}$$

Since we are now dealing with partial differential equations, the previous constants $A$ and $B$ are allowed to depend on $V$.

Given these solutions, we can evaluate the Casimir:

$$R = V^2 - |J|^2 = V^2 - A(V)^2 - B(V)^2. \tag{21}$$

If it equals zero, we have $A(V)^2 + B(V)^2 = V^2$, and there is a $P_0$ such that $A(V)/V = -\sin(\delta P_0)$ and $B(V)/V = -\cos(\delta P_0)$. Thus,

$$\text{Im} J(V, P) = V \sin(\delta(P - P_0)) \tag{22}$$

$$\text{Re} J(V, P) = V \cos(\delta(P - P_0)) \tag{23}$$

or:

$$J(V, P) = V \exp(i\delta(P - P_0)). \tag{24}$$

The canonical realization of (1), given by Casimir–Darboux coordinates $(V, P, R)$, identifies $J$ as a "holonomy modification" of the classical Hamiltonian (14), in which the Hubble parameter

represented by the momentum $P$ is replaced by a periodic function of $P$.[1] The vanishing Casimir, $R = 0$, then appears as a reality condition for $P$ in (24).

We conclude that the paradigmatic bounce model of loop quantum cosmology, analyzed numerically in [19], is a canonical realization of harmonic cosmology.

## 3. Group Field Theory as a Bosonic Realization of Harmonic Cosmology

The canonical realization constructed in the preceding section is faithful: the number of Darboux coordinates agrees with the rank of the Poisson tensor given by (1), and the number of Casimir coordinates agrees with the co-rank. If one drops the condition of faithfulness, inequivalent realizations can be constructed which even locally are not related to the original system by canonical transformations. We will call "realization equivalent" any two systems that are realizations of the same model. This notion of equivalence therefore generalizes canonical equivalence. As we will show now, this generalization is crucial in relating loop quantum cosmology to group field theory.

### 3.1. Bosonic Realizations

Instead of canonical realizations, one may consider bosonic realizations, replacing canonical variables, $(q, p)$ such that $\{q, p\} = 1$, with classical versions of creation and annihilation operators, $(z, \bar{z})$ such that $\{\bar{z}, z\} = i$. The map $z = 2^{-1/2}(q + ip)$ defines a bijection between canonical and bosonic realizations.

The brackets (1) correspond to the Lie algebra su(1, 1). A different real form of this algebra, sp(2, $\mathbb{R}$), has a large number of (non-faithful) bosonic realizations given by the special case of $N = 1$ in the family of realizations:

$$A_{ab}^{(n)} = \sum_{\alpha=1}^{n} \bar{z}_{a\alpha} \bar{z}_{b\alpha} \quad , \quad B_{ab}^{(n)} = \sum_{\alpha=1}^{n} z_{a\alpha} z_{b\alpha} \quad , \quad C_{ab}^{(n)} = \frac{1}{2} \sum_{\alpha=1}^{n} (\bar{z}_{a\alpha} z_{b\alpha} + z_{b\alpha} \bar{z}_{a\alpha}) \tag{25}$$

of sp(2N, $\mathbb{R}$) [20–23] with relations:

$$[A_{ab}, A_{a'b'}] = 0 = [B_{ab}, B_{a'b'}] \tag{26}$$

$$[B_{ab}, A_{a'b'}] = C_{b'b}\delta_{aa'} + C_{a'v}\delta_{ab'} + C_{b'a}\delta_{ba'} + C_{aa'}\delta_{bb'} \tag{27}$$

$$[C_{ab}, A_{a'b'}] = A_{ab'}\delta_{ba'} + A_{aa'}\delta_{bb'} \tag{28}$$

$$[C_{ab}, B_{a'b'}] = -B_{bb'}\delta_{aa'} - B_{ba'}\delta_{ab'} \tag{29}$$

$$[C_{ab}, C_{a'b'}] = C_{ab'}\delta_{a'b} - C_{a'b}\delta_{ab'} . \tag{30}$$

The indices take values in the ranges $\alpha = 1, \ldots, n$ and $a, b = 1, \ldots, N$, where $a \leq b$ in $A_{ab}$ and $B_{ab}$. There are $2nN$ real degrees of freedom in the bosonic coordinates $z_{a\alpha}$, while sp(2N, $\mathbb{R}$) has dimension $N(2N + 1)$.

For $N = 1$, we have three generators:

$$A^{(n)} = \sum_{\alpha=1}^{n} \bar{z}_\alpha \bar{z}_\alpha \quad , \quad B^{(n)} = \sum_{\alpha=1}^{n} z_\alpha z_\alpha \quad , \quad C^{(n)} = \frac{1}{2} \sum_{\alpha=1}^{n} (\bar{z}_\alpha z_\alpha + z_\alpha \bar{z}_\alpha) \tag{31}$$

---

[1]　In loop quantum cosmology [15], the Hubble parameter is obtained from a component of the Ashtekar connection, which, as in the full theory of loop quantum gravity, is not represented directly as an operator, but only indirectly through holonomies [16,17]. In isotropic models, matrix elements of holonomies along straight lines are of the form $\exp(i\delta P)$, as it appears in (24). In loop quantum cosmology, going back to [18], the parameter $\delta$ has sometimes been related to the area spectrum in loop quantum gravity. However, this relationship is ad-hoc, and therefore, it is not surprising that no such role of $\delta$ can be seen in the present realization.

with relations:

$$[A^{(n)}, B^{(n)}] = C^{(n)} \quad , \quad [A^{(n)}, C^{(n)}] = -2A^{(n)} \quad , \quad [B^{(n)}, C^{(n)}] = 2B^{(n)}. \tag{32}$$

For any $n$, the identification:

$$A^{(n)} = i\bar{J}/\delta \quad , \quad B^{(n)} = iJ/\delta \quad , \quad C^{(n)} = 2iV/\delta \tag{33}$$

relates these brackets to (1).

### 3.2. Model of Group Field Theory

In [24], a toy model of group field theory has been derived that produces bouncing cosmological dynamics for the number observable of certain microscopic degrees of freedom. Starting with a tetrahedron, the model assigns annihilation and creation operators to the sides, which change the area in discrete increments. For an isotropic model, the four areas should be identical, and their minimal non-zero value is determined by a quantum number $j = 1/2$, modeling the discrete nature through a spin system following the loop paradigm [25]. Each isotropic excitation has the "single-particle" Hilbert space $(1/2)^{\otimes 4}$, which contains a unique spin-two subspace. Since this subspace consists of totally-symmetric products of the individual states, it is preferred by the condition of isotropy. Restriction to the spin-two subspace then implies a five-dimensional single-particle Hilbert space with complex-valued bosonic variables $A_i$.

A simple non-trivial dynamics is then proposed [24] by the action:

$$S = \int \mathrm{d}\phi \left( \frac{1}{2} i \left( A_i^* \frac{\mathrm{d}A^i}{\mathrm{d}\phi} - \frac{\mathrm{d}A_i^*}{\mathrm{d}\phi} A^i \right) - \mathcal{H}(A^i, A_j^*) \right) \tag{34}$$

in internal time $\phi$. The first term indeed implies bosonic Poisson brackets $\{A_i^*, A^j\} = i\delta_i^j$. The second term is fixed by proposing a squeezing Hamiltonian:

$$\mathcal{H}(A^i, A_j^*) = \frac{1}{2} i\lambda \left( A_i^* A_j^* g^{ij} - A^i A^j g_{ij} \right) \tag{35}$$

with a coupling constant $\lambda$ and a constant metric $g_{ij}$ with inverse $g^{ij}$. The metric is defined through an identification of the spin-two index $i$ with all totally-symmetric combinations of four indices $B_I \in \{1, 2\}$ taking two values, such that:

$$g_{(B_1 B_2 B_3 B_4)(C_1 C_2 C_3 C_4)} = \epsilon_{(B_1(C_1} \epsilon_{B_2 C_2} \epsilon_{B_3 C_3} \epsilon_{B_4)C_4)} \tag{36}$$

with separate total symmetrizations of $\{B_1, B_2, B_3, B_4\}$ and $\{C_1, C_2, C_3, C_4\}$, respectively, and the usual totally antisymmetric $\epsilon_{BC}$. Ordering index combinations as:

$$i \in (1, 2, 3, 4, 5) = (1111, (1112), (1122), (1222), (2222)), \tag{37}$$

the metric can be determined explicitly as the matrix:

$$g = \begin{pmatrix} 0 & 0 & 0 & 0 & 1 \\ 0 & 0 & 0 & -1 & 0 \\ 0 & 0 & 1 & 0 & 0 \\ 0 & -1 & 0 & 0 & 0 \\ 1 & 0 & 0 & 0 & 0 \end{pmatrix}. \tag{38}$$

A second crucial observable, in addition to the Hamiltonian, is the excitation number,

$$V = \frac{1}{2}\left(A_i^* A^i + A^i A_i^*\right),$$
(39)

identified with the cosmological volume following group field cosmology. This volume evolves in internal time $\phi$ according to the Hamiltonian $\mathcal{H}$. Solutions for $V(\phi)$, derived in [24], show bouncing behavior (4) that can be modeled by the effective Friedmann equation (10).

We can now readily show that this behavior is not a coincidence: The metric (38) has eigenvalues $+1$ with three-fold degeneracy and $-1$ with two-fold degeneracy. Diagonalizing it by an orthogonal matrix gives linear combinations $z_\alpha$ of the $A^i$ and $A_i^*$ that preserve the bosonic bracket $\{A_i^*, A^j\} = i\delta_i^j$, defining a bosonic transformation:

$$z_1 = \frac{1}{\sqrt{2}}(A^1 + A^5) \quad, \quad z_2 = \frac{1}{\sqrt{2}}(A^2 - A^4) \quad, \quad Z_3 = A^3$$
(40)

for eigenvalue $+1$, and:

$$z_4 = \frac{1}{\sqrt{2}}(A^1 - A^5) \quad, \quad z_5 = \frac{1}{\sqrt{2}}(A^2 + A^4)$$
(41)

for eigenvalue $-1$.

We can deal with the negative eigenvalues in two ways. First, multiplication of $z_4$ and $z_5$ with $i$ preserves the bosonic bracket and leads to a metric $g_{ij}' = \delta_{ij}$. We then have $\mathcal{H} = \frac{1}{2}i\lambda(A^{(5)} - B^{(5)})$ for (35) and $V = C^{(5)}$ for (39). Alternatively, using only diagonalization by an orthogonal matrix, we have:

$$\mathcal{H} = \frac{1}{2}i\lambda\left(A^{(3)} - B^{(3)} - (A^{(2)} - B^{(2)})\right)$$
(42)

and:

$$V = C^{(3)} + C^{(2)}$$
(43)

where $z_1$, $z_2$, and $z_3$ contribute to the $n = 3$ realization and $z_4$ and $z_5$ to $n = 2$. Observing (33) and the fact that the relations (1) are invariant under changing the sign of $J$, the volumes and Hamiltonians in both loop quantum cosmology and group field theory are identified with the same generators in harmonic cosmology. The models of loop quantum cosmology and group field theory are therefore realization equivalent.

## 4. Implications and Further Directions

There is an immediate application of our result to the appearance of singularities in the model [24] of group field cosmology. As argued in this paper, because the volume is derived from the positive number operator of microscopic excitations $A^i$, it can be zero only at a local minimum, which requires $V(\phi_{\min}) = 0$ and $dV/d\phi = 0$ at some internal time $\phi_{\min}$. The combination of these two conditions is quite restrictive, and [24] concludes that a singularity (zero volume) can be reached only for a small number of initial conditions.

However, our identification of the model of [24] as a bosonic realization of harmonic cosmology suggests a more cautious approach to the singularity problem. In su$(1, 1)$, there is no positivity condition on the generator that corresponds to the volume $V$. The bosonic realization in terms of microscopic excitations $A^i$ is therefore local, in the sense that the $A^i$ are local coordinates on the Poisson manifold that realizes harmonic cosmology, and $V = 0$ is at the boundary of a local chart. Accompanying $V(\phi_{\min}) = 0$ by $dV/d\phi = 0$ is therefore unjustified unless one can show that evolution never leaves a local chart. The condition $V(\phi_{\min}) = 0$ is not as restrictive as the combination, and it leaves more room for solutions that reach zero volume (These solutions may still be considered non-singular if there is a unique Hamiltonian that evolves solutions through zero volume.

In loop quantum cosmology, evolving through $V = 0$ is interpreted as changing the orientation of space [15,26].).

In harmonic cosmology, further generalizations of the model used here have already been explored in some detail. The new relationship with group field theory suggests similar generalizations also on the group-field side of the equivalence. For instance, harmonic cosmology can be defined for any power-law $Q = a^p$ replacing $V = a^3$, describing a quantization ambiguity that corresponds to lattice refinement of an underlying discrete geometry [27,28]. The same algebra, with arbitrary exponent $p$, can then be realized bosonically, suggesting related group-field models (while the power-law $V = a^3$ is preferred at large volume because it avoids an expansion of the discrete scale to macroscopic size, a different power-law may well be relevant near a spacelike singularity).

Another parameter related to the relation $V = a^3$ is the averaging volume $V_0$ used to define the isotropic model. We have implicitly assumed $V_0 = 1$ in order to focus on algebraic properties; in general, we have $V = V_0 a^3$ where $V_0$ is computed as the coordinate volume of the averaging region. Classical equations do not depend on $V_0$, but quantum corrections do, as can be seen here from the fact that in the action (34), the Hamiltonian $\mathcal{H}$ is proportional to $V_0$, but the symplectic term is not. The microscopic action is then not invariant under changing $V_0$. The implications of a relation between $V_0$ and the infrared scale of an underlying field theory [29] are of importance for the interpretation of quantum cosmology [30], and similar conclusions should hold true in group-field cosmology.

In classical harmonic cosmology, the Casimir $R = 0$ is exactly zero, but this value usually changes in the presence of quantum corrections [1,2,31]. The bouncing behavior (2) is no longer guaranteed if $R < 0$ and $|R| > (\delta H_\varphi^\delta)^2$, because $V(\phi)$ behaves like a sinh under these conditions. These conditions require large quantum corrections, greater than the matter density related to $p_\phi^2$. They are therefore unlikely to be fulfilled in a macroscopic universe. However, as pointed out in [30], an appeal to the BKLscenario [32] near a spacelike singularity shows that a homogeneous model is a good approximation only if it has small co-moving volume, given by the averaging volume $V_0$ mentioned above. Such a tiny region does not contain much matter energy, which can then easily be surpassed by quantum corrections in a high-curvature regime: $p_\phi \propto V_0$ is suppressed for small $V_0$, while volume fluctuations $\Delta V$ are not proportional to $V_0$ because they are bounded from below by the $V_0$-independent $\hbar$ in uncertainty relations. The genericness of bouncing solutions in loop quantum cosmology or group-field cosmology is then not guaranteed.

Finally, a large class of microscopic models can be constructed from the bosonic realizations of harmonic cosmology with arbitrary $n$ in (31) (the fact that group field cosmology leads to non-faithful realizations of harmonic cosmology with a potentially large number of microscopic degrees of freedom is a consequence of the "second quantization" made use of in group field theory). The question of whether these are related to group field cosmology in some way appears to be of interest.

**Author Contributions:** Conceptualization, M.B.; writing—original draft preparation, M.B.; formal analysis, B.B., M.B. and S.C.; writing—review and editing, B.B., M.B. and S.C.

**Funding:** This research was funded by NSF Grant Number PHY-1607414.

**Conflicts of Interest:** The authors declare no conflict of interest.

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
