# Peer review of "Equivalence of Models in Loop Quantum Cosmology and Group Field Theory"

_universe, doi:10.3390/universe5020041_

Round 1

Reviewer 1 Report

This paper demonstrates the mathematical equivalence of a recently proposed toy model of GFT cosmology (see ref. [20]) with harmonic cosmology (see ref. [1,2]), which is a representation of loop quantum cosmology (LQC) in terms of harmonic oscillators. The paper is very well written, and clarifies the mathematical origin of some unexplained coincidences between LQC and GFT that were pointed out in [20]. My recommendation is to publish the paper without major revision in its current form. I only have one comment or remark, which may be addressed by the authors if they would find it necessary/interesting: GFT is often presented as a second quantisation of LQG, harmonic cosmology, on the other hand, is an example of a conventional first-quantised theory. The article demonstrates that the two are mathematically equivalent. It seems, therefore, that the notion of first or second quantization is rather meaningless in this context, and this observation could be addressed (or refuted) in a short sentence in the conclusion.

Author Response

This is an interesting suggestion, thank you. In fact, the distinction between first and second quantization is recognizable in the realizations we discuss, because a second quantization leads to new degrees of freedom which imply non-faithful realizations in the corresponding models. We have inserted a parenthesis in the last paragraph of our last section to point this out.

Reviewer 2 Report

This is an interesting paper in which the authors point out that LQC (Loop Quantum

Cosmology) and GFT (Group Field Theory) cosmology can be viewed as different

representations of the same algebra of operators. I consider this to be an important

result.

I suggest that the authors try to make the paper a bit more accessible. The present

version starts out with the algebra of operators (1), without mentioning what these

operators are. It becomes clear later on that V is the volume, and the imaginary

part of J becomes the Hamiltonian. But the physical meaning of the complete J

(in particular its real part) is not clear to me. If the authors could start out their

introductory section by discussing the physical meaning of the operators it would

help readers like myself a lot. There is a discussion of this issue after (24), but it

is a bit abstract.

In LQC, the constant $\delta$ which already appears in the definition of the Hamiltonian

is related to the quantization of the area operator. How does one see this

meaning emerging from the abstract algebra representation? If the authors could

comment on this it would also be nice.

I also find the paragraph below (24) hard to read. Can the authors explain a bit more?

In particular, maybe explain a bit without making use of the LQC jargon.

On Page 5 I noticed potentially confusing notation: on the previous page $j$ is an

integer, but on this page a half integer $j$ appears.

Conflict of interest statement: ``author'' should be pluralized.

Author Response

Thank you for these suggestions. The first paragraph of the paper now explains the physical meaning of the variables in more detail. In particular, the real part of J is not independent of the imaginary part and V if the Casimir is fixed (to zero, in our application). Therefore, it does not have its own physical interpretation.

The relationship between delta and the area spectrum in some formulations of loop quantum cosmology has not been derived from first principles. Since we do not make assumptions beyond the algebraic structure and the dynamics, no such relationship can be seen in our realizations. Footnote 1 on page 4 now points this out. The same footnote provides more details of loop quantization and holonomy modifications in cosmological models.

We have changed the notation, using indices a,b instead of i,j in equations (25)-(30) to avoid confusion with j=1/2 used on the next page, and we have corrected additional typos.

Reviewer 3 Report

The authors explore the equivalence between loop quantum cosmology, an established field of quantum cosmology and group field cosmology, as emerging sub field which has closer connection to the full quantum gravity theory. The work is clearly presented alongwith assumptions and caveats. I think the manuscript is a valuable step in this direction and definitely meets the standards of the journal. So, I am happy to recommend this paper to be published without any further modification. A spell check would be great as there are some minor typos in the paper.

Author Response

Thank you for this evaluation. We have found and corrected several typos.